## Overview Review

biodiversity loss; planned extinction; human-driven extinction; ethics and policy; de-extinction

**Corresponding author:**
Anna Wienhues;
Email: anna.wienhues@ifikk.uio.no

# The ethics of species extinctions

Anna Wienhues[1] , Patrik Baard[1] , Alfonso Donoso[2] and Markku Oksanen[3]

[1]Department of Philosophy, Classics, History of Art and Ideas, University of Oslo, Oslo, Norway; [2]Institute of Applied Ethics and Institute of Political Science, Pontificia Universidad Católica de Chile, Santiago, Chile and [3]Department of Social Sciences, University of Eastern Finland, Kuopio, Finland

## Abstract

This review provides an overview of the ethics of extinctions with a focus on the Western analytical environmental ethics literature. It thereby gives special attention to the possible philosophical grounds for Michael Soulé's assertion that the untimely 'extinction of populations and species is bad'. Illustrating such debates in environmental ethics, the guiding question for this review concerns why – or when – anthropogenic extinctions are bad or wrong, which also includes the question of when that might not be the case (i.e. which extinctions are even desirable). After providing an explanation of the disciplinary perspective taken (section "Introduction"), the concept of extinction and its history within that literature are introduced (section "Understanding extinction"). Then, in section "Why (or when) might anthropogenic extinctions be morally problematic?", different reasons for why anthropogenic extinctions might be morally problematic are presented based on the loss of species' value, harm to nonhuman individuals, the loss of valuable biological variety and duties to future generations. This section concludes by also considering cases where anthropogenic extinctions might be justified. Section "How to respond to extinctions?" then addresses a selection of topics concerning risks and de-extinction technologies. Finally, the section on "Extinction studies" introduces other viewpoints on the ethics of extinction from the extinction studies literature, followed by the "Conclusion".

## Impact statement

This is an overview review article of the ethics of species extinctions drawing on the environmental ethics literature. While most people seem to believe that extinctions are morally bad or wrong, no systematic review of the moral philosophical literature that can support or question such intuitions on this subject matter has been provided to date, which is the gap that this review article aims to fill.

## Introduction

In an influential essay, conservation biologist Michael Soulé offered a list of what he termed 'postulates of conservation biology'. Amongst these were also 'normative postulates' (i.e. value statements) such as that '[d]iversity of organisms is good' and, relatedly, that 'the untimely extinction of populations and species is bad' (1985, 730, italics in original removed). Soulé did not present any detailed argumentative justification for these normative postulates (see Baard, 2022, 20ff), yet much could and must be said about their moral grounds, which provides an invitation to further discuss the ethical basis of species extinctions. So, *why* exactly are extinctions bad or wrong?

Extinctions – either as the extinction of single species or as mass extinction events on earth – are not only a matter of scientific inquiry and of policy relevance, but also a subject addressed by *moral philosophy* through questions concerning, amongst other things, the moral relevance and ontological status of species, moral responsibilities for human-caused extinctions, any moral obligations to avoid the loss of species and so on. Thus, this body of literature will aid us to explore the question of why extinctions might be bad or wrong, and for that purpose, we will provide a review of philosophical articles and books pertaining to the ethics of extinction (setting aside adjacent fields such as environmental law and history).

Moral philosophy, or ethics, is a normative discipline distinguished from empirical research on people's values, preferences and practices as done in sociology, economics and history. Moral philosophy aims to assess critically and systematically the normative reasons that are expressed through/in value judgements, decisions and actions. It ranges from interrelated investigations about normative and evaluative languages and the relationship between fact and value (often known as meta-ethics), to research about fundamental ethical principles and ethical theories such as utilitarianism, deontology and virtue ethics (normative ethics), to the moral scrutiny of real-

world cases and problems (practical/application-oriented/area-specific ethics). These three areas are often intertwined, and what is often called 'applied ethics' is not about merely applying more general or fundamental ethical theories. Rather, application-oriented practical reasoning gives rise to novel challenges and insights, which in turn also influence more general and fundamental discussions in ethics.

*Environmental ethics* spans across all these three areas of moral philosophy with a particular focus on concepts and arguments relating to the environment and nature, including the ethics of extinction, which gives rise to discussions about values and duties but also about issues concerning concepts such as species, ecosystems or biodiversity. Due to problems of deriving an action-guiding 'ought' from a descriptive 'is',[1] ethical discussions are pivotal to reasoning about and acting upon environmental issues.

Based on this disciplinary perspective, different positions and arguments presented here are not necessarily compatible with each other, because there are disagreements on different argumentative premises, intuitions, sentiments and underlying ethical theories. Therefore, in the following we are presenting different argumentative positions, but *not* a philosophical consensus on the ethical dimensions of species extinctions, such as in the form of a coherent theory that most authors would agree on, which is similar to how there are also disagreements in the natural and social sciences regarding which theory provides the best explanation of a natural or social phenomenon, which is not a weakness of this literature but an inherent feature of philosophical discussions that aim to critically exchange and improve arguments and theories. Ultimately, the moral judgements one finds reasonable or justified regarding extinction will not only depend on one's empirical premises (and the quality thereof) but also on different metaphysical, ontological and normative premises and intuitions about how one sees the natural world and how one relates to it and lives within it.

Illustrating such debates in environmental ethics, the guiding question for this review concerns *why* – or *when* – anthropogenic (i.e. human-caused) extinctions are bad or wrong, which also includes the question of when that might not be the case (i.e. which extinctions are even desirable). This overarching question guides this review in two ways. Firstly, it led our selection of relevant positions and areas of discussion to be included in the review. In that regard, we aimed for as much comprehensiveness as possible pertaining to the representation of the main categories of reasons that have been presented in the field that can argumentatively support (or question) Soulé's postulate. Secondly, this question narrows the scope of this review by including only arguments and positions that pertain to the ethics of extinction as opposed to providing a review of the much broader field of conservation ethics (about how to protect nonhuman individuals, species, ecosystems and landscapes). Consequently, we will cover, on the one hand, the major areas of discussion regarding the ethics of extinction (such as a growing body of literature on de-extinction technologies) and, on the other hand, we only include authors that provide substantive arguments on the topic of extinction (narrowly conceived).

To illustrate the breadth of such considerations, we will provide an overview of different concerns and arguments that are being discussed in the Western anglophone environmental ethics literature on species extinctions, broadly within the tradition of analytical philosophy. So, two caveats ought to be mentioned with respect

to the chosen literature for this overview and the narrative presented. Firstly, because this review is largely limited to the Western anglophone analytical literature (except for some of the authors mentioned in the last section on extinction studies, which is necessitated by the character of that area of research), it can neither represent the rich variety of arguments and positions found in other philosophical traditions such as so-called continental philosophy and non-Western philosophy (e.g. Asian or African environmental philosophical traditions and Indigenous philosophies), nor does it cover relevant philosophical literature published in other languages. Secondly, by starting with Soulé's postulate this review paints a particular narrative of the relevant reasons speaking in its favour or against it. We thereby particularly emphasise matters of moral value (i.e. different intrinsic and instrumental values) for providing reasons against or in favour of species extinctions due to their centrality in the environmental ethics literature, as well as in interdisciplinary conservation debates more broadly. Yet, certainly other somewhat different narratives that, for example, highlight more the differences between different ethical theories (e.g. rights-based versus virtue-focused perspectives) would be as plausible.

In the following, we start by presenting some noteworthy themes that have emerged from discussions within environmental ethics that address the concept of extinction and its history within that literature (section "Understanding extinction"). Then, we turn to different reasons for why anthropogenic extinctions might be morally problematic and cases where anthropogenic extinctions might be justified (section "Why (or when) might anthropogenic extinctions be morally problematic?") and how to respond to extinctions (section "How to respond to extinctions?"). Finally, we look beyond our own narrow disciplinary perspective and introduce other viewpoints on the ethics of extinction from a body of work called 'extinction studies' (section "Extinction studies") before concluding (section "Conclusion").

## Understanding extinction

### The history of the concern about extinction in environmental ethics

A period from the late eighteenth century to the early twentieth century changed the way extinction was thought about in Europe and North America. During that time, three developments came together. Firstly, mostly thanks to Darwin, the idea of evolution and extinction as its key component replaced creationist ideas that distrusted the very possibility of extinction, if it was not outright denied. Reasons for this former denial are historical, as the influential Greek philosophers Plato and Aristotle, unlike some popular writers of the time, paid no attention to fossil bones of no longer existing vertebrates (Mayor, 2011). For them, species could not disappear except locally – a belief in line with the principle of plenitude dating back to Plato (Lovejoy, 1936), which states that all possible lifeforms necessarily exist and cannot disappear. Figuratively, the extant kinds form a 'great chain of being' and each kind in this chain is there necessarily; thus, the chain is unbreakable, and nature is in balance. This idea prevailed in pre- and early modern Europe and was endorsed by, for instance, Linnaeus as late as the eighteenth century.

Secondly, given the importance of the notion of extinction to evolutionary thinking, extinction was not seen as a matter of concern but rather the opposite: extinct species were too weak to survive, or an extinct species was replaced by another, thus maintaining the balance in nature. Both cases – through weeding out the

---

[1]This is often referred to as 'Hume's law' according to which prescriptive conclusions do not follow logically from descriptive premises.

weak and introducing novel species – were regarded as improvements in natural evolution (Barrow, 2009; Sepkoski, 2020).

Thirdly, despite this, and most relevant for thinking about the *ethics* of extinction, a concern about extinctions as universal and irreversible events started taking shape (as early as Marsh, 1864/2003) and was broadly reflected in new legislation such as the Sea Birds Preservation Act of 1869 in the UK. This historical change is well conveyed by Aldo Leopold (1949/1989, 110), when he stated that '[t]he sailor who clubbed the last auk thought nothing at all. But we, who have lost our [passenger] pigeons, mourn the loss'. Leopold (1989, 210–211) also already noted that economic considerations fail to provide reasons for protecting those species that lack economic value. Instead of extending economic valuation, as has been proposed later by economists, Leopold saw conservation as action against extinction that also rests on non-anthropocentric ethical ideas. Nevertheless, a systematic philosophical discussion on the ethics of extinction evolved much later with the formation of the academic field of environmental ethics since the 1970s.

Early books and articles on environmental ethics did not explicitly discuss the ethics of extinction. The discussion focused on the flipside of the coin: Why protect endangered species (Elliot, 1980; Gunn, 1980; Rescher, 1980)? One could see here a connection with new legislation, such as the US Endangered Species Act (1973), but the discussion broadened to also include the significance of diversity in nature more broadly (Naess, 1973; Norton, 1987), to incorporate the possibility of multiple extinctions in a relatively short period of time (for mass extinction, see Sepkoski, 2020) and to introduce a new concept, biodiversity (Wilson, 1986). For example, some ethicists have tried to develop a naturalistic account of the human propensity to value species and their diversity (Callicott, 1984, 305, 1986; or, more recently, elaborated E.O. Wilson's biophilia hypothesis, Baxter, 2007; cf. Takacs, 1996).

Arguably, one of the earliest contributions on the ethics of extinction is Richard Sylvan's (then still known as Routley) famous thought experiment of the *last person on earth* (originally the 'last man'), firstly formulated in 1973, which can be made relevant for thinking about extinction. Sylvan writes:

> The last man (or person) surviving the collapse of the world system lays about him, eliminating, as far as he can, every living thing, animal or plant (but painlessly if you like, as at the best abattoirs). What he does is quite permissible according to basic chauvinism, but on environmental grounds what he does is wrong (Sylvan, 1973/2003, 49).

Later in the same article, he also specifically considers 'vanishing species':

> Consider the blue whale (…). The blue whale is on the verge of extinction because of his qualities as a private good, as a source of valuable oil and meat. The catching and marketing of blue whales does not harm the whalers; it does not harm or physically interfere with others in any good sense, though it may upset them and they may be prepared to compensate the whalers if they desist; nor need whale hunting be willful destruction. (…) The behavior of the whalers in eliminating this magnificent species of whale is accordingly quite permissible – at least according to basic chauvinism. But on an environmental ethic it is not (1973/2003, 50).

He asked whether species or nature more broadly are valuable beyond their potential usefulness for humans. What Sylvan's famous case suggests is that the permissibility of extinguishing species – as he argued, allowed by traditional Western ethical systems that conceive of nature as something to be used by humans as they please – is an example of what he called 'human chauvinism', now better known as 'anthropocentrism' (for a critical discussion of the 'last person' argument, see Peterson and Sandin, 2013). Sylvan's thought experiment has successively been modified and discussed by many other philosophers such as Mary Ann Warren (1983) and Robin Attfield (1981, 1983). The latter engaged in this type of argument to scrutinise the possibility of human extinction in the context of nuclear armament and 'ecocatastrophe'. He aimed to show that non-sentient living nature such as trees has morally relevant interests and that causing their extinction by the 'last man' would be morally wrong, even if it would not affect any other humans.

Writing at the same time as Sylvan, Joel Feinberg (1974) addressed the ethics of species protection from a different argumentative angle. Because he argued that only (human and nonhuman) individuals with interests can have rights, he rejected the idea that species have a right to exist (for an opposing view, see Staples and Cafaro, 2012). Therefore, he claimed that humans have a duty to protect endangered species, but the ultimate addressees of this duty are future humans, not the species themselves (see also Elliot, 1980). Likewise, Nicholas Rescher (1980) depicted species conservation 'as a quintessential humanitarian task' and John Passmore (1974) considered wanton eradication of species to be acts of vandalism.

In brief, very early on, there were opposing opinions about *why* human-caused extinctions are morally wrong, while there was largely agreement on the issue *that* anthropogenic extinctions are wrong (with exceptions such as the smallpox virus). The similarity between these different views was that they regard extinction as a loss of value, either for humanity or for the universe as considered from a human-independent perspective. Independently of whether one endorses an anthropocentric (i.e. human-focused) or non-anthropocentric rationale for species conservation, both give rise to a range of resulting questions that were formulated in the literature of environmental ethics early on, such as: Are species individual entities or life processes that come to an end with extinction (Rolston, 1985, 1995) (section "Defining extinction")? What are acceptable costs for protection efforts (Norton, 1987; Naess, 1989), and relatedly, should all species, even the most harmful ones, be conserved (Rolston, 1985; Johnson, 1991) (section "Future generations")? Are species as collectives more valuable than nonhuman individuals (Russow, 1981; Johnson, 1991) and does that therefore justify the culling of individuals that threaten the existence of native populations (Warren, 1997; Varner, 1998)?

### Defining extinction

What constitutes an extinction is a complex subject matter. However, for the purpose of ordinary concerns about extinction and for how moral philosophers discuss its moral relevance, most theorists' primary concern lies with *anthropogenic species extinctions* – while treating 'extinctions' and 'species extinctions' synonymously. Moreover, while many theorists put an emphasis on extinctions on a *mass scale* (such as Bendik-Keymer, 2014; Cafaro, 2015; Panagiotarakou, 2016; Baquedano Jer, 2019), some theorists are also explicitly concerned with the extinction of *individual* species (such as Rolston, 1995).[2] Accordingly, there are four conceptual components found in many accounts of the ethics of extinction.

Firstly, there are different ontological positions on the meaning of '*species*', that is what a species is (see Delord, 2007; Maclaurin and Sterelny, 2008; Ereshefsky, 2022), which then also have different

---

[2]A different subject that we set aside concerns anti-natalist perspectives that argue in favour of *human* extinction (e.g. Lenman, 2002).

ethical implications (such as in Hull, 1978; Gruen, 2011; Smith, 2016). Moreover, because different kinds of species conservation practices change the genetic makeup of the target species (via conventional breeding programmes or synthetic biology) and/or influence their behaviour, it is also necessary to consider (1) whether the resulting population still belongs to the same species (which in turn influences whether we consider it extinct or not) and (2) whether that matters morally (see Preston, 2021). Thus, there are different conceptions of species and these impact how extinction is defined (see Maclaurin and Sterelny, 2008; Godfrey-Smith, 2014; Korsgaard, 2018a).

Secondly, the term '*extinction*' can also be understood in different ways (see Delord, 2007; Tanswell, 2022), which is of ethical relevance because (1) it frames what falls *within* the ethics of extinction and, in turn, (2) the way extinction is constructed in controversial cases is *in itself* subject to ethical scrutiny. When speaking about extinctions as something morally problematic, many (not all) authors will refer to 'final' extinctions, if not specified further in works of practical ethics. Thus, an emphasis is put on 'global' (all individuals of a species are gone) as opposed to 'local' (loss of a population) extinctions. However, because a species can be extinct in several senses of the term, some authors also specifically problematise, for example cases where there still exist some (or even many) living individuals belonging to the species of concern. That is, it is not finally extinct, but it may be 'functionally' extinct. Drawing on the example of the whitebark pine as a case of a functional extinction, Christopher Preston (2021, 4, italics in original) argues in this regard that '[t]he concern with this type of extinction is not that you will never see individuals of the taxon again. The concern centres on the *anthropogenic reordering of relationships.* Not a loss of life, but a loss of arrangement'. That is, regarding this type of extinction an emphasis is put on changing ecological relationships rather than on the circumstance whether some individual members of the species are still alive.

Thirdly, many positions will argue – in different ways – that there is a moral difference between *anthropogenic* and *non-anthropogenic* extinctions. In Holmes Rolston's III words, '[n]atural extinction opens doors, anthropogenic extinction closes them' (2012, 139). Independently from whether Rolston's view on the matter is convincing, some people will intuitively find it morally objectionable that a species in the form of all the individuals belonging to that species abruptly disappears due to human actions (such as in Sylvan's thought experiment, see section "The history of the concern about extinction in environmental ethics") and thereby also ending an evolutionary process, instead of taking issue with natural evolutionary processes in themselves. This distinction not only assumes that anthropogenic and non-anthropogenic extinctions are conceptually different (Aitken, 1998), but usually also relies on the judgement that human-caused extinctions are morally more problematic than non-anthropogenic extinctions. So, for many authors it is relevant to know whether human causation is at play for the ethical assessment of the extinction in question (for epistemic barriers to assessing this causation, see Tanswell, 2022) and that, in turn, is closely related to judgements about *moral responsibility* (Oksanen, 2007). Contrasting views consider all species extinctions to be similarly morally problematic (i.e. similar 'badness') and thereby downplay the moral relevance of anthropogenic causation by adding 'wrongness' (Powell, 2011).

The fourth conceptual distinction needs to be made between *mass* extinctions and *non-mass* extinctions, with mass extinctions differing from an ordinary extinction event in several ways (see

Bendik-Keymer and Haufe, 2016). Because of these differences, evaluative judgements about the current 'sixth mass extinction' event on earth seem to be often based on the intuition that mass extinctions are morally more problematic than single extinction events. For example, consider how different terms such as 'environmental atrocities' (see Card, 2004) or 'ecocide' (see Baquedano Jer, 2019) can be potentially linked to extinction. While many might think that single extinction events added together are as bad as a mass extinction event involving the same number of species lost, some authors argue that a mass extinction is worse – for example due to additionally involving a 'planetary shift' (Sandler, 2021a).

## Why (or when) might anthropogenic extinctions be morally problematic?

Returning to our question from the introduction, conservation biologists Soulé and Wilcox famously claimed that '[d]eath is one thing; an end to birth is something else' (1980, 8). This implies an evaluative judgement about the badness of species extinctions, but what might that mean? In the more recent environmental ethics literature, we can find a variety of reasons given to explain why anthropogenic extinctions might be (or might not be) morally problematic. We will present the main (but not sole) reasons found in this literature focusing on (1) the value of species, (2) harming nonhuman individuals, (3) the value of biodiversity and (4) duties to future (human and nonhuman) generations. In the last part of this section, we will also turn to (5) instances where extinctions might even be morally desirable.

### Involving loss of value due to the loss of species

Because species extinctions involve the loss of species, the most obvious place to start is to think about whether this loss is *in itself* problematic (see Passmore, 1974). A common argumentative route in this regard is to argue that species are valuable and, thus, their extinction involves a loss of something that has been important to someone. For example, certain species might be *instrumentally* valuable because they are useful for human purposes (see section "Leading to the loss of valuable biological variety"). However, species might also be valuable independently from their actual or potential use value. That is, in contrast they might have *intrinsic* value as entities that we value in themselves.[3] Let us briefly introduce three (non-exhaustive) versions of this position (for further examples, see Callicott, 1986 [for a critique, see Lo, 2001]; Gorke, 2003; for a more general critical analysis of holistic accounts of species value, see Agar, 2001).

For one, authors such as Rolston have argued that species – not understood as classes but as 'living historical forms' (Rolston, 1985, 2012) – are objectively intrinsically valuable ('some values are objectively there—discovered, not generated, by the valuer'. 1988,

---

[3]What is understood as 'intrinsic value' can vary considerably between different authors. For good overviews, see O'Neill (1992) and McShane (2007). Thus, not all authors will agree on classifying the accounts that we introduce in the following as positions *on* intrinsic value (e.g. as opposed to extrinsic value). In addition to instrumental and intrinsic values, a third category of value that has been proposed is '*relational value*', which, in turn, has also been applied to species (see Deplazes-Zemp and Chapman, 2021). Also, this latter type of value is interpreted very differently by different authors (for a critique, see Baard, 2019, 2022), but arguments about relational value can be related to some of the concerns about the loss of species that are discussed in section "Extinction studies".

116), which means that with each species that disappears, we lose something of value even if we fail to recognise this loss. In his words,

> [t]he species defends a particular form of life, pursuing a pathway through the world, resisting death (extinction), by regeneration maintaining a normative identity over time. It is as logical to say that the individual is the species' way of propagating itself as to say that the embryo or egg is the individual's way of propagating itself. The value resides in the dynamic form; the individual inherits this, exemplifies it, and passes it on (1994, 21).

However, only anthropogenic extinctions are subject to moral evaluation (a duty to avoid extinctions) in Rolston's account. He relies for this purpose on the analogy between 'natural death' (i.e. a natural extinction) and 'murder' (an anthropogenic extinction), which leads him to assess each anthropogenic extinction as a 'superkilling' (Rolston, 1995). That is, on Rolston's account the metaphor of a superkilling illustrates the badness of extinctions by putting an end to an evolutionary 'story' and precluding its future evolutionary possibilities. Other authors such as Lawrence E. Johnson have spelled out similar ideas, but with less emphasis on evolution and rather in terms of the species' interests and their resulting moral standing (see Johnson, 1991).

Criticism of accounts such as Rolston's or Johnson's position, in turn, has been aimed at different aspects of their theories. Two primary areas of debate stand out. Firstly, the plausibility of the underlying theory of objective intrinsic value in Rolston's account is a matter of debate (for a critique, see Elliot, 1980; Callicott, 1992). Against the background that intrinsic values were intensely discussed in the field of environmental ethics at the time when Rolston published his account (see O'Neill, 1992), it should be noted that other authors such as Callicott (1984, 1986) defended alternative value accounts relying on subjective values or human sentiments as the basis of a non-anthropocentric environmental ethic. Secondly, Johnson's position relies on the idea that species can be considered to be akin to living beings (i.e. it relies on a specific species concept – as a living entity – that has been widely criticised not only by individualist perspectives but also by other holistic accounts; see Smith, 2016) and, thus, also be morally considerable in the same way. However, whether that analogy is convincing and whether species have morally relevant interests are also a matter of debate (for a critique, see Sandler and Crane, 2006; Sandler, 2012). Consequently, in the more recent literature other arguments in favour of the moral value of species that are not based on such an understanding of species are more widespread.

For example, secondly, Ian Smith argues in favour of the view that a species has an 'intrinsic good' that 'consist in its abilities to flourish', based on a Hennigian species concept that sees species as 'historical individuals', which, in turn, gives us reasons to preserve species on Smith's account because, amongst other things, '[a species'] flourishing is a species' organisms continuing to reproduce successfully (that is, the organisms producing fertile offspring) and the species remaining safe from extinction' (2016, 14). Smith takes it for granted that 'species taxa are real' (2016, 3–4) but subspecies are not. Therefore, only species are of intrinsic value. Critics have seen this presumption, in turn, as a failure to properly consider subspecies and higher categories in taxonomy (Burbrink et al., 2022).

Finally, Ronald Sandler understands species as 'groups of biologically related organisms that are distinguished from other organisms by virtue of their shared *form of life*' (Sandler, 2012, 6, italics in original). What distinguishes organisms in this way, according to Sandler, are, for example, their ways to acquire energy (what they eat), how they move (and whether they move at all), how they reproduce and so on. Because Sandler argues against objective values being applicable to species (understood in this way), he points out that species can be still valuable in other ways. For example, some species are subjectively (finally/non-instrumentally) valuable (i.e. valuable because people actually value them) in two ways: firstly, in the form of 'preference value' as the 'value that something has because people have a preference for it' (2012, 23). That value is also sometimes called 'existence value', which according to Espen Stabell (2019, 180) 'can be understood to involve self-regarding as well as other-regarding and nonanthropocentric preferences'. Secondly, species can also have 'integral value' as the 'value that something possesses when it is valued in a way that flows from one's worldview of core value commitments' according to Sandler (2012, 24).

This last account of the value of species neither relies on species having objective value (like Rolston), nor on species having interests akin to living beings (like Johnson), nor on the premise that species are being able to flourish (like Smith), and thus, it relies on less premises that might be controversial. However, in contrast to those other accounts its scope is much narrower because it concedes that only *some* anthropogenic species extinctions are lamentable by involving the loss of such value and only if this value is actually held by some humans.

### Resulting from harm to nonhuman individuals

Non-anthropocentric *individualist* accounts within environmental ethics – that is *sentientist* (i.e. all sentient beings are morally considerable; Singer, 1975) or *biocentric* (i.e. all living beings are morally considerable; Taylor, 1986/2011) accounts – are only indirectly concerned with extinctions. What primarily matters to such positions is either the harm, impediment to well-being or injustice that is borne by individual nonhuman beings, and thus, the relevant harm involved in extinctions is not found on the species level (Nussbaum, 2011). Yet, it is often argued – for different reasons – that these two levels of analysis are closely related (Agar, 2001; Regan, 2004; Taylor, 2011; Korsgaard, 2018a; Donoso, 2019; Baard, 2022). For example, according to animal rights theorist Tom Regan the preservation of the biotic community follows from showing 'proper respect for the rights of the individuals' (Regan, 1983/2004, 363) that are part of that community.

As an example of emphasising the current anthropogenic mass extinction within an individualist framework, Anna Wienhues (2020) argues in favour of a biocentric theory of interspecies justice. Within this account, anthropogenic extinctions function as indicators of potential previous distributive injustices to nonhuman beings, due to the unjust deprivation of habitat that can ultimately lead to species extinctions. Within this context, extinctions are not injustices in themselves, but an outcome of interspecies injustice – metaphorically speaking she argues that 'each anthropogenic extinction is a *memorial for past injustices*' (2020, 157, italics in original).

A commonality of all of these individualist accounts is that they are only indirectly concerned with extinctions and cannot explain why an extinction is *in itself* morally lamentable (Benton, 1993). Moreover, such accounts usually cannot morally distinguish between saving a specimen that belongs to a near-extinct species and one that does not belong to such a species (Sober, 1986/1995; Baard, 2022). If ethical relevance is placed solely on individual entities rather than species, then the extinction status of the species

the entity belongs to does not necessarily count ethically, which could be more of a problem for some individualist accounts (e.g. Singer, 1975; Cochrane, 2012; see Korsgaard, 2018b), than for individualist accounts that allow for a degree of value pluralism (as opposed to everything of moral relevance being reducible to, e.g., individual well-being) and/or that allow for a degree of ethical pluralism (e.g. Wienhues, 2020).

An alternative solution is presented by individualist accounts that conceptualise the individual's interests as extending to 'all other organisms that are relatives' in terms of its species (understood following Mayr's biological species concept) (Agar, 2001, 150). Based on Nicholas Agar's (2001) account, that consequently allows for an individualist argument that proposes that individuals that are part of endangered species and therefore rare are more valuable than their plentiful counterparts. According to Agar that follows because the value of an individual is not 'a function only of its own goals. Its demise affects the other-directed and other-requiring goals of its conspecifics, depending on how plentiful its species is' (Agar, 2001, 150), which includes, for example, the ability of individuals to mate and successfully reproduce.

Yet, non-individualist accounts consider a theoretical focus on the individual to be a weakness. As Johnson puts it, '[a]n ethic that deals adequately with the issues of extinction must not only avoid being anthropocentric, it must avoid being atomistic' (1991, 170, italics in original). That might make one favour again one of the earlier introduced values of species accounts, but an account like Johnson's faces in turn a range of difficulties in its own as already indicated above.

### Leading to the loss of valuable biological variety

Besides accounts that consider the loss of *parts* of biodiversity to be problematic because of the moral relevance of species or nonhuman individuals, other perspectives consider species extinctions to be problematic due to the involved loss of biological *variety*. While biodiversity loss does not necessarily follow from species extinctions and vice versa, they are clearly conceptually related and practically implicated in each other. Problematising the loss of (bio)diversity, in turn, is a notable topic in the context of species extinctions because it has been institutionalised on an international level via the Convention of Biological Diversity (established in 1993).

Although the convention text begins with the recognition of 'the intrinsic value of biological diversity' (CBD, 1993), it is not obvious what role the intrinsic value in the CBD plays, as it appears only in the preamble. The recognition of intrinsic value is also followed by stating that the parties are conscious of 'the ecological, genetic, social, economic, scientific, educational, cultural, recreational and aesthetic values of biological diversity' (CBD, 1993). This shows that there is ambiguity in how the value of biodiversity is to be understood. Moreover, the larger normative international legal framework and even the CBD itself mainly endorse the established resource-focused language and practice (see Oksanen and Vuorisalo, 2019). Here, again the question arises about how biodiversity-focused claims about value should be interpreted.

So, the extinction of species via the resulting loss of biodiversity seems to involve a loss of value, but reasons for why the loss of variety might be problematic can rest on different argumentative grounds (Oksanen, 1997; Callicott, 2017; Heinzerling, 2017; McShane, 2017; Odenbaugh, 2017; Baard, 2022). Moreover, biodiversity can be conceptualised in various ways (CBD, 1993; Norton, 2008; Faith, 2017, 2021; Sarkar, 2017; Burch-Brown and

Archer, 2017) and normative emphasis can be put on different kinds of diversity (such as 'phylogenetic diversity', Palmer and Fischer, 2021). Generally, there are two common (not exhaustive) argumentative routes to choose from, which do not preclude each other but are often found in combination to justify the protection of biodiversity.

Firstly, biological *variety* as in the variety or abundance of, for example, genes or species might have *instrumental value* beyond the (instrumental) value of the species themselves (e.g. Maclaurin and Sterelny, 2008). While that is a common premise in conservation biology and environmental ethics that can justify why the extinction of many species is problematic due to the loss of valuable variety, different argumentative grounds for such instrumental value such as biodiversity's contribution to ecosystem functioning or its (potential) agricultural and pharmaceutical benefits have also been critically discussed (see Maier, 2012; Deliège and Neuteleers, 2015; Newman et al., 2017; Morrow, 2023). Independent of such critiques, the loss of variety might not be problematic for instrumental value reasons in all circumstances (e.g. consider the loss of variety of certain pathogens), which means that *not necessarily all* extinctions involve a loss of instrumentally valuable variety.

Yet that needs to be kept distinct from the instrumental value attributed to species themselves (such as their 'incalculable instrumental value', see Smith, 2022), although not all species will be instrumentally valuable for human purposes either (see section "Future generations"). In so far as these, as well as biological variety, are fundamental grounds for the possibility of human life on earth, a mass extinction can be understood as humanity's 'autodestruction' constituting 'a *wrong in its own category*. The wrong is the wrong of putting an end to humankind and all that is of value in it. This is a *cataclysmic wrong*' as put by Bendik-Keymer and Haufe (2016, 7, italics in original). Thus, for such an argument the 'massness' (see section "Defining extinction") of the current extinction event is central and, thereby, it avoids the problem that some individual species might not be instrumentally valuable for human purposes.

The second option is to consider the *intrinsic value* of biological variety as something that might be valuable in itself – independently of whether it is beneficial for humans or other lifeforms (Mikkelson, 2022). There are different versions of what this entails (see McShane, 2017). For example, deep ecologist Arne Naess famously stated that '[r]ichness and diversity of life forms (…) are also values in themselves' (Naess, 1986/2003, 264). Providing a more nuanced view on the intrinsic value of biodiversity, Gregory Mikkelson (2011, 186) points out that 'variety' is not the sole determinant of value in neither Næss' deep ecology, nor in Mikkelson's own preferred richness theory. Both invoke additional determinants such as 'harmony' (Mikkelson, 2011, 2014). To Mikkelson, more related to extinction, 'anthropogenic species extinctions are decreasing the overall harmony of life on Earth' (2011, 191).

Philosophical claims about biodiversity's intrinsic value also get external support. As already mentioned, conservation biologist Soulé considered biotic diversity to have intrinsic value (1985, 731) and also the Convention on Biological Diversity refers to intrinsic value (for a critical discussion of both, see Baard, 2022). Moreover, studies about the environmental values of the public also report intrinsic values being supported (e.g. Leiserowitz et al., 2005; discussed by both Mikkelson, 2014; Odenbaugh, 2017) and strong pro-environmental attitudes confirmed empirically have the possibility of generating subjectivist non-instrumental values (see Lo, 2006). However, different proposals concerning the intrinsic value

of biological variety have also been subject to critical analysis (Sarkar, 2005; Kraut, 2011; Maier, 2012; Newman et al., 2017), such as concerning whether intrinsic value can attach to biological variety itself or whether that is better understood in terms of intrinsically valuing parts of biological variety such as species and individual beings (McShane, 2017).

Moreover, in addition to valuing species, nonhuman individuals and biological variety, further (additional or alternative) adjacent reasons grounding the badness of species extinction can be found in the literature. To mention two examples broadly within the realm of biodiversity-focused arguments, that includes, for one, arguments pertaining valuing the '*irreplaceable design* that is embodied by the individuals' part of a species (Cline, 2020, 46, italics in original). The point being that after a long evolutionary process, '[e]ach of the wonderful contrivances of nature are (…) unique and irreplaceable. Such irreplaceable design has special value that we recognise when we experience awe and wonderment towards the sophisticated structures and systems and strategies exemplified by living organisms' (Cline, 2020, 59). According to Brendan Cline, that shifts the focus of concern away from the species themselves towards such 'irreplaceable design'. Secondly, the loss of biodiversity might also be considered problematic due to a lack of respect towards nonhuman living beings and active nature in their *natural otherness.* Such respect provides us with reasons to protect biodiversity and, consequently, species variety (Wienhues and Deplazes-Zemp, 2022). Subsequently, human-caused extinctions can be interpreted as a lack of such respect.

What is common to all these types of biodiversity-focused arguments – regarding the instrumental value of variety, the intrinsic value of variety and pertaining alternative concepts (harmony, design, otherness, etc.) – is that for them the focus of moral concern shifts from species extinctions towards biological variety or those other concepts. So, like the individualist arguments (section "Resulting from harm to nonhuman individuals") extinctions are again not problematised because of the loss of species themselves, but rather due to what this loss means for other valued entities or other matters of moral concern.

### Future generations

A further reason for why species extinctions might be morally problematic concerns our duties to *future human generations* (see Feinberg, 1974; Sandler, 2021b). In essence, if we believe that the interests of future generations put current generations under an obligation to not frustrate these interests, we have a reason to protect species and biodiversity that future generations might (intrinsically and instrumentally) value and, thereby, avoid 'irreversible loss' (Spiekermann, 2022). While there might be considerable uncertainty about what species, ecosystems and/or ecosystem 'services' future generations will actually value and/or need, this ultimately involves again an assessment of different arguments pertaining to the value of species and biodiversity (as seen in sections "Involving loss of value due to the loss of species" and "Leading to the loss of valuable biological variety"). Moreover, considerations about duties towards future generations in terms of species conservation provide a bridge between the ethics of extinction and the ethics of sustainability (see Norton, 2003; Armstrong, 2021).

In addition to these human-focused arguments, we can also consider *non-anthropocentric arguments pertaining to the future* (building on some of the arguments found in sections "Resulting from harm to nonhuman individuals" and "Resulting from harm to nonhuman individuals"; see also Nolt, 2021; Whyte, 2022). There

are two main argumentative routes open in this regard. Firstly, one can consider potential duties towards future nonhuman beings or species in so far as the extinction of other species will affect them negatively (and consider cases where species might benefit from the extinction of other species; related to this, see Palmer, 2011 and Cripps, 2013 for reflections about how certain species will benefit from climate change). Like how in intra-human intergenerational ethics a lot of thought has been given to the 'nonidentity problem' (see Norton, 1982; Parfit, 1986), also here we face a nonhuman nonidentity problem (Palmer, 2011), in so far as anthropogenic species extinctions will affect which individuals will come into existence in the future. That is, the extinction of some species will have a knock-on effect on which individuals of still remaining species will meet and mate, which in turn will influence which individuals will live in the future. The underlying philosophical problem is that those future individuals would not have come into existence without the extinction (which, in turn, is relevant for underlying considerations about how the 'harm' of extinction is understood). Secondly, the case of extinctions also leads to a 'nonexistence problem' (Wienhues, 2020) in so far as nonhuman individuals, species and ecosystems in the future will not only be affected by prior extinctions, but many nonhuman individuals will simply never come into existence.

### Can anthropogenic extinctions ever be justified?

Some readers might assume that *all* anthropogenic extinctions are morally problematic, but does that judgement hold up in all cases? Consider these two examples from species conservation practice. Firstly, concerning 'local' (as opposed to 'global') extinctions consider restoration-based reasons in favour of the extinction of a population of an '*invasive' species* (for critical discussions, see Rawles, 2004; Odenbaugh, 2022; Thresher, 2022). Secondly, other conservation practices such as *captive breeding and release* programmes have in turn been criticised on the grounds of not allowing for a 'death with dignity' (Chessa, 2005). While in this case the extinction might still seem morally problematic, such programmes might not present an appropriate response. In both cases, it is possible to claim that an extinction can be all-things-considered better than an alternative course of action, for instance, on ecocentric or sentientist argumentative grounds (which both, in turn, can also be engaged to criticise the moral desirability of such extinctions).

Also, consider the case of *disease-transmitting species* such as certain mosquitoes that transmit malaria, dengue fever, zika and so on, which is the most discussed case of potentially justified extinction. The global health burden of these diseases is very high (WHO, 2017; Greisman et al., 2019), which provides a good reason to consider whether the eradication of such disease-transmitting species would be an appropriate means to address this problem (Fang, 2010; Bates, 2016). While it is an empirical question to decide whether this constitutes an approach that is likely to succeed while minimising the risks in comparison with alternatives, this also generates a range of ethical questions to consider, such as regarding the moral standing of individual nonhuman animals, ethical aspects of risk and the intrinsic and/or instrumental value of species (see Pugh, 2016; Wienhues, 2021; Callies and Rohwer, 2022). A particular emphasis found in the literature on bioethics concerns questions of research ethics, risk and consent regarding genetically engineering species such as mosquitoes in this context (see Resnik, 2014; Emerson et al., 2017; Resnik, 2017; Neuhaus and Caplan, 2017; Patrão Neves and Druml, 2017; Meghani and Boëte, 2018). Notably, even some authors who consider

anthropogenic extinctions to be highly morally problematic acknowledge exceptions for disease-transmitting species such as malaria-transmitting mosquitoes (e.g. Rolston, 2001; Smith, 2016).[4] While the use of synthetic biology such as gene drives remains controversial (Preston and Wickson, 2019), the health burden of vector-borne diseases constitutes a weighty reason to take them into consideration.

Intended species extinctions are also considered for other – even more controversial – purposes, which include the case of predation raised in animal ethics. While some animal ethicists consider wild animal suffering to be of serious moral concern (Horta, 2017), a few authors go as far as arguing that there are reasons speaking in favour of the *extinction of predatory species*.[5] The benefit hoped for would be the reduction of (non-anthropogenic) suffering in the 'wild' caused by predation (see McMahan, 2010, 2016). Such proposals to alleviate 'wild' animal suffering (by a range of means aimed at 'dewilding' or 'redesigning nature', see Duclos, 2022; Kianpour and Paez, 2022) are in turn not shared by other animal and environmental ethicists on more general terms (see Palmer, 2010, 2015; Donaldson and Kymlicka, 2011; Aaltola, 2014; Hettinger, 2018), as well as specifically on the question of removing predatory species (Vincelette, 2022). Reasons include the difficulties of implementing such proposals and more principled objections towards such interventions into 'nature'.

## How to respond to extinctions?

Assuming that anthropogenic extinctions are morally problematic, how should they be avoided? What are morally permissible (or even demanded) means of doing so? One way of morally responding to extinctions is to turn our attention to species that have not yet gone extinct, but which are at *risk* of extinction (related to this, see arguments about the relevance of *rarity* [e.g. Gunn, 1980; James, 2024] and arguments that emphasise *endangered* species [e.g. Smith, 2016]).

This relates to the much broader body of work on the *conservation of species* in which new means and policies are being developed and discussed from a multidisciplinary perspective. To give one example (out of many) of philosophical discussion pertaining endangered species, consider 'assisted migration' or 'species translocation'. The discussion about such proposals originated in grassroot activism but was taken up by ecologists, and ethicists soon joined the debate (Sandler, 2010, 2012; Hällfors et al., 2014; Palmer and Larson, 2014; Siipi and Ahteensuu, 2016; Minteer, 2017; Preston, 2018; Palmer, 2021). The idea is to transfer populations to more suitable habitats where those species did not exist before to prevent them from going extinct. This is a conservation approach that is meant to be appropriate because of climate change (Hällfors et al., 2014). There are different views on this matter. For example, there are discussions about how intentional translocation affects the value of the target species and its habitats (Siipi and Ahteensuu, 2016; Siipi, 2017). To Sandler, who focuses on the ethics of species, assisted colonisation is only justified in rare occasions in which 'the translocated species

would be ecologically or instrumentally valuable in the recipient site (and this can be reliably predicted in advance), or the species has final value' (2012, 85) (see section "Involving loss of value due to the loss of species" regarding Sandler's understanding of such value). From an animal rights perspective, in turn, Angie Pepper argues that 'the permissibility of practices like assisted migration hinges on whether or not we can justify imposing risks on animals in order to prevent rights violations further down the line' (2019, 602).

## Risk of extinction

For the purposes of moral theorising, 'risk' refers to the likelihood of a loss or harm taking place, which makes 'risk' a value concept referring to an unwanted outcome through human actions, in addition to an epistemological dimension where that outcome is likely to happen with a specific probability (Hansson, 2013). If that probability is unknown, it is referred to as 'uncertainty' rather than 'risk', which often motivates a precautionary approach. With respect to species extinction, there are two main ways of discussing risks and extinctions: the first focuses on the risk of a single species facing extinction, and the second pays attention to broader systemic and accumulative risks, either in terms of mass extinction or in terms of a collapse of the biospheric system as we know it.

Regarding single species, risk is a premeditative concept that enables us to respond to near-extinction, such as when a species extinction is predicted to happen in the light of best ecological modelling, where the predicted quantity of extinctions has often been referred to as 'extinction debt' (Hanski and Ovaskainen, 2002; Kuussaari et al., 2009). Against this background, deliberations on employing concepts and ideas drawn from ecological sciences and (meta)population biology are taking place within environmental ethics and the philosophy of biology, which includes topics of ethical relevance such as decision-making under uncertainty and the applicability of the precautionary principle, land-use planning, protected area networking and prioritisation (Norton, 1987; Sarkar, 2005).

Risks have value dimensions, and what value is justifiably ascribed to a species will play a part in how it is to be prioritised (see section "Why (or when) might anthropogenic extinctions be morally problematic?"). Most efforts to save near-extinct species require *prioritising*: global and national Red Lists of Threatened Species are examples of such a conservation policy, as is Bryan Norton's 'triage model' (1987, 258). Yet, prioritisations come with a range of philosophical problems. According to Norton, many of those challenges arise due to the difficulties of reducing the values of species to a single scale (1987, 255).

A broader perspective to the risk landscape emerges when we consider what kinds of systemic risks each extinction gives rise to (which is related to some of the concerns introduced in section "Leading to the loss of valuable biological variety"). A common argument rests on an analogy between species extinctions and the loss of rivets on an airplane wing (Ehrlich and Ehrlich, 1981).[6] It aims to show that the ethical significance of extinction is due to its potential effects on ecosystems (the loss of too many

---

[4]Distinct but related to these concerns are broader considerations about the 'ethics of pests' (see Draney, 1997; Winston, 1999). Even if there are good moral reasons for anthropogenic extinctions, the full-scale eradication of a so-called 'pest' can be infeasible and have unintended consequences in practice. Therefore, some authors such as Mark Winston (1999) consider management and control better options than eradication.

[5]Within the animal ethics literature, another additional type of concern relevant for theorising extinctions can be found, which are approaches that argue in favour of the abolition of the use of domesticated animals and, therefore, the extinction of these kinds of animals (e.g. Albersmeier, 2014).

[6]Another kind of systemic argument relies on the 'option value' of species (see Maclaurin and Sterelny, 2008; Newman et al., 2017), which is a utilitarian concept in which ethical relevance is placed on the overall aggregated outcomes of a decision. In practice, an emphasis on the option value of species could allow for their substitution, if there are other ways of reaching the same results by other means than saving a near-extinct species. Such potential substitutability is problematic, in turn, for a range of ethical views, such as accounts that attribute intrinsic value to species (see section "Involving loss of value due to the loss of species").

'rivets'), which in turn are prerequisites for human and nonhuman survival. When conjunct with uncertainty regarding the number of rivets, or species, that an airplane wing, or ecosystem, can dispose of without collapsing, it becomes essential to preserve species and avoid their extinction (for a critical discussion of the analogy, see Sarkar, 2005).

This expresses a form of *precautionary principle*, which has been discussed both critically and approvingly from different philosophical perspectives (Sandin, 2004; Sarkar, 2005; Hourdequin, 2007; Newman et al., 2017; Tanswell, 2022). The precautionary principle is also a part of the preamble of the Convention of Biological Diversity (CBD, 1993) where it is stated that lack of full knowledge ought not to justify inaction, and it is mentioned in the framework of the Intergovernmental Science-Policy Platform on Biodiversity and Ecosystem Services (IPBES n.d.). The complexity and uncertainty pertaining to how ecosystems work (Regan et al., 2002; Sarkar, 2005) provide support for a precautionary principle concerning environmental decision-making. Yet, some authors consider its policy implications problematic due to leading to logical contradictions by potentially recommending two opposing actions (e.g. recommending the development of and simultaneously the prevention of developing genetically modified organisms), failing to account for possible benefits, neglecting that precaution has costs and for lacking conceptual clarity (Newman et al., 2017; critiqued in turn in Baard, 2022).

### De-extinction

What should be done after extinctions have already occurred? Besides more conventional species conservation practices, there is a broader discussion on the ethics of developing and using biotechnologies for conservation purposes and beyond (Ehrenfeld, 2006; Basl and Sandler, 2013; Kaebnick and Murray, 2013; Oksanen and Siipi, 2014; Preston, 2018). Setting these larger concerns aside, more technology-focused conservation approaches – if justifiable in certain cases – should be seen as complementary rather than alternatives to conventional conservation practices (Sandler, 2021b). Accordingly, the subject of debate concerns whether there are certain cases where such technological means are appropriate and desirable at all.

For example, a topic that has recently been gaining attention is the evaluation of the possibility of engineering species (via gene drives, cloning, etc.) either with the purpose of avoiding their extinction, such as by genetically adapting them to new climatic conditions or by increasing the genetic diversity of a population (see discussions by Palmer, 2016; Rohwer, 2018; Sandler, 2019; Preston, 2021; Sandler, 2021b; Welchman, 2021), or as means to eradicate invasive species (Thresher, 2022).

Related to proposals in favour (or critiques) of altering species to avoid (or cause) their extinction, a second body of work focuses on how and whether methods such as back-breeding, cross-species cloning and genetic engineering can be employed to appropriately address extinctions *once* they have occurred. Thus, the possibility of *de-extinction* by bringing a species 'back' from extinction has captured the imagination of many environmental philosophers (amongst many others, see Cohen, 2014; Oksanen and Siipi, 2014; Turner, 2014; Diehm, 2015; Minteer, 2015; Campbell and Whittle, 2017; Kasperbauer, 2017; Oksanen and Vuorisalo, 2017; Sandler, 2017; Browning, 2018; Preston, 2018; Katz, 2022; Sandler et al., 2022; for an alternative narrative, see also Odenbaugh's, 2023 recent review of the ethics of de-extinction).

This topic has become the centre of a lively debate in the ethics of extinction, which is noteworthy in so far as de-extinction is only a minor subject within the broader field of conservation ethics and debates (see also section "Extinction studies"). Of concern are, for example, different moral values and goods that might be supported or hindered by de-extinction projects (e.g. Cohen, 2014; Haught, 2017; Smith, 2017; Welchman, 2017; Rohwer and Marris, 2018), which also include anthropocentric reasons in favour of the de-extinction of certain species, such as their potential value for ecosystem functioning, their economic value or their scientific value.

Non-anthropocentric attempts to provide a qualified justification for this approach include providing a list of criteria for its employment (e.g. Kasperbauer, 2017) or by trying to make the idea useful for other normative purposes such as whether de-extinction might be a way of enacting a reparative duty towards nonhuman species or individual animals after extinctions have occurred. While thinking about de-extinction in terms of reparation or restitution might be intuitively appealing and constitute an argument occasionally engaged by conservation professionals, concerns about de-extinction as a form of reparation or retribution are shared by many authors (such as Oksanen and Siipi, 2014; Diehm, 2015; Lean, 2020; Welchman, 2021; yet for an argument in favour of a prima facie obligation to re-create species, see Jebari, 2016), while more might be said in favour of de-extinction as ecological restoration (Turner, 2014). Overall, while some authors argue that de-extinction can be ethically permissible in certain circumstances (Sandler, 2019), the debate on the desirability of de-extinction has been rather sceptical. Three common (not exhaustive) critiques of de-extinction fall into the following three categories.

Firstly, it has been frequently pointed out that the resulting species is not the same species as the previously extinct species, but only constitutes a copy (which concerns the species' 'identity', Blockstein, 2017; Siipi and Finkelman, 2017; Lean, 2020). Actual 'resurrection' is not considered to be possible – or only possible by drawing on a very specific species concept (Oksanen and Siipi, 2014) and related metaphysical commitments to resolve the 'resurrection paradox' (Delord, 2014) – which leaves any de-extinction proposal wanting (Diehm, 2017) and reintroduces the abovementioned considerations about the ontology of species (Beever, 2017).[7] However, other writers argue that de-extinction technologies should not be understood as aiming to re-create extinct species in the first place (Lean, 2020), which weakens this critique aimed at de-extinction while also reducing its intuitive appeal.

Secondly, concerns about animal welfare have been raised (Gamborg, 2014; Oksanen and Siipi, 2014; Kasperbauer, 2017; Turner, 2017; Browning, 2018; Browning and Veit, 2022), which regards, for example, the welfare of the animals used for laboratory testing, of the animals that are (re)introduced into ecosystems and of the animals already living in the target ecosystems. Distinct from but related to such worries are also concerns regarding the ecological risks of (re)introducing species into ecosystems and, thus, what this means for ecosystems beyond the de-extinct species in question (Oksanen and Siipi, 2014; on the more general relationship between biodiversity conservation and resurrecting species, see Oksanen, 2014).

Thirdly, we can find a range of worries raised against de-extinction on a more abstract level of analysis concerning the human–nature relationship (Kohl, 2017; Katz, 2022). That is

---

[7]It should be noted that these are not necessarily new philosophical questions. For example, in 1994 Robert Elliot already asked whether it is possible to recreate an extinct species.

linked, in turn, to worries about naturalness (Gamborg, 2014) in terms of artificiality and authenticity (Katz, 2022; discussed in Campbell, 2022; Lean, 2022; Preston, 2022; Reydon, 2022; Turner, 2022, but for an argument against this concern, see Campbell and Whittle, 2017) as well as concerns about instrumentalist 'techno-optimist' conservation strategies more generally as exemplified by de-extinction (Diehm, 2015).

## Extinction studies

In the previous sections, we have mainly focused on what is commonly termed an analytical moral philosophical approach to the ethics of extinction. Yet, several other philosophical perspectives on extinction and contributions from outside of the fields of moral philosophy and the philosophy of biology also provide insights that are pertinent for thinking about the ethics of extinction that complement and enrich some of the discussions considered throughout this article. For example, there is a vibrant sub-debate in the conservation literature according to which ethical concepts and values should be supplemented or replaced by aesthetic concepts and values (e.g. Passmore, 1974; Sober, 1986/1995; Hargrove, 1989; Thompson, 1995; Tribot et al., 2018). Thus, one could claim that a species extinction is analogous to the loss of a great work of art.

A particularly notable further area of debate is the philosophical contributions in the interdisciplinary overlap with the environmental humanities and social sciences (e.g. Barad, 2007; Haraway, 2008; Sodikoff, 2012; Tsing, 2015; Guasco, 2021) in a broad research area called 'extinction studies' (e.g. Chrulew et al., 2017). Before concluding, we will highlight some elements of this perspective to find areas of convergence and divergence between these different intellectual traditions working on the ethics of extinction.

Scholars working within the extinction studies framework tend to emphasise an understanding of extinctions as a biocultural phenomenon (van Dooren et al., 2017; van Dooren, 2018) that cuts across reality, inviting a vision of existence that recognises the deep interwoven nature of multispecies communities, experiences and collaborations. In line with this understanding, some writers have conceptualised extinctions as the loss of ways of life, underlining the fact that extinctions represent the demise of distinct forms of mating, nurturing, intergenerational learning, educating, interacting and so on (Rose, 2011; Crist, 2013; van Dooren, 2014; Despret, 2017; Hatley, 2017).

Influenced by the work of philosophers such as Edmund Husserl, Maurice Merleau-Ponty and others, this perspective emphasises a phenomenological approximation to extinction, opposing a mere behavioural approach to the nonhuman world and, instead, stressing the importance of attentiveness and lived experience in our interactions with the natural world (Lestel et al., 2014). A key point of this approximation is to underline the experiences of those affected by extinction rather than focusing on the abstract disappearance of species. This speaks to the idea that the very experience of living on a biologically impoverished earth is understood as a source of deep concern (Crist, 2019). This methodological emphasis, in turn, motivates the interest of extinction studies in individual stories of extinction (e.g. van Dooren, 2014) investigated through the tools of so-called field philosophy (Buchanan et al., 2018) and other alternative methods (e.g. Masco, 2017).

Another important aspect of this perspective on analysing extinctions is the idea that the consideration of processes and occurrences of extinction cannot be separated from a critique of various forms of colonialism, imperialism and capitalism that inform our extinction-related practices and concepts (Dawson, 2016; Salazar Parreñas, 2018). To illustrate, returning to themes introduced in section "Defining extinction", while extinction studies do not attribute a single, all-encompassing meaning to extinction (De Vos, 2007), this phenomenon is regarded as the result, amongst other things, of complications derived from concepts used hegemonically, including the terms 'species', 'biodiversity', 'extinctions' and 'scientific knowledge', amongst other notions. It is argued that these concepts represent a conventional classification of the natural world that is tinted by colonialist and anthropocentric biases that universalise Western scientific perspectives, thus privilege some organisational hierarchies over others and naturalise distinct power relations within dominant biodiversity conservation paradigms (Theriault et al., 2020). These terms employed with the purpose of organising reality 'exclude myriad forms of life and relations and draw sharp boundaries between "living" and "dead" that confound the basic principles of so many living cosmologies' (Mitchell, 2016b).

When thinking about responses to extinction – for example as an 'ethos of responsiveness towards the phenomenon of mass extinction' (Michell, 2016a, 39) – also in extinction studies, there is a general scepticism towards de-extinction projects as shared by the approaches introduced above (see section "De-extinction"). If extinctions are understood not as the demise of the last individual of a population, but as the disappearance of a way of life, de-extinction projects are doomed to fail, because they do not seem capable of capturing the deeply relational nature of life (Friese, 2013; Jørgensen, 2013). This is interestingly complementary to some of the views discussed in the previous sections that, when examining the wrongness of extinctions and the problems of technological solutions such as de-extinction, emphasise the relevance of ecological and cultural relationships as the basis for what has value. Confronted with the failure of preventing extinctions and the magnitude of the biocultural losses produced by the disappearance of these ways of life, some extinction studies scholars emphasise the importance of mourning and remembrance as a proper response to extinction (van Dooren and Rose, 2017; de Massol de Rebetz, 2020). Rather than undoing loss, the moral significance of reflecting – remembering, mourning, grieving and imagining – the disappearances of deeply interwoven ways of life is underlined, giving rise to a moral psychology aspect to extinction.

## Conclusion

This overview of the ethics of species extinctions has not attempted to spell out or critically discuss all topics or argumentative positions in detail. More humbly, our aim has been to provide an initial glimpse into some of the themes relevant for thinking about extinction that are being discussed in the literature of environmental ethics. For that purpose, we have given special attention to the possible philosophical grounds for Michael Soulé's assertion that the untimely 'extinction of populations and species is bad', which does not answer what should be done to prevent extinctions.

While there is no simple or straightforward answer to the question of *why* anthropogenic extinctions are 'bad' or morally wrong, there is clearly largely agreement on the issue *that* anthropogenic extinctions are in most cases wrong, except for the controversially discussed cases of planned species extinctions in support of human (and nonhuman) health, such as with respect to vector-borne diseases. In such cases, anthropogenic extinctions might even be morally required. Besides such special cases, there is a broad

plurality of arguments and perspectives underlying and supporting the need to pursue expansive biological conservation policies. *Which* conservation strategies and practices should be pursued (and which should not), in turn, remains an additional matter of debate, the subject of the ethics of conservation that supplements the ethics of extinction.

Lastly, there are two remarkable aspects to the literature surveyed. Firstly, most arguments pertaining to the wrongness or badness of species extinctions are not primarily concerned with 'species' as such. Rather, the matter of moral concern, such as the locus or source of value, is often located somewhere else: in the nonhuman individuals involved, in people's relationship to particular species or landscapes, in biodiversity including variability within species and between ecosystems and so on, which reflects that there is more of moral relevance to the ethics of extinction than merely the numerical loss of species and their respective intrinsic or instrumental value. Views on the moral wrongness of anthropogenic extinctions result also, amongst other things, from underlying premises of what constitutes an appropriate human–nature relationship, which is why the current anthropogenic mass extinction event is of significant moral concern.

Secondly, considering how marginal de-extinction technologies still are as a (potential) practice of biodiversity conservation, it is noticeable how much interest the environmental ethics literature has shown on this topic. However, more important than the interrelated questions of whether species can and should be 'resurrected' is the uptake of a second set of broader related questions. These questions concern the morally permissible (or impermissible) uses of biotechnology for conservation purposes under conditions of climate change when conventional conservation practices such as habitat preservation and even more proactive approaches such as assisted migration reach their limits of effectiveness. As this is a growing area of interdisciplinary debate, we believe that providing critical analysis of these questions in the intersection between environmental ethics and the ethics of technology will be a significant area of environmental philosophical contribution during the upcoming years.

**Open peer review.** To view the open peer review materials for this article, please visit http://doi.org/10.1017/ext.2023.21.

**Acknowledgements.** The authors would like to thank Timo Vuorisalo and three anonymous reviewers for their helpful comments on a previous draft of this manuscript.

**Author contribution.** A.W. is the lead author. P.B., A.D. and M.O. have all contributed in equal amounts to the writing and revising of this review.

**Financial support.** The writing of this review was supported for AW by the Forschungskredit of the University of Zurich, grant no. FK-21-081, and AW has also received funding from the European Research Council (ERC) under the European Union's Horizon 2020 research and innovation programme (grant agreement no. 948964, project 'Dynamic Territory'). For PB, this article has received funding from the European Research Council (ERC) under the European Union's Horizon 2020 research and innovation programme (grant agreement no. 948964, project 'Dynamic Territory'). AD's work was supported by the Fondecyt Project No. 11160170 and the ANID Millennium Science Initiative – ICS2019_025.

**Competing interest.** The authors declare none.

**Social media summary.** This review provides an overview of the ethics of extinctions drawing on the environmental ethics literature. It thereby gives special attention to the possible philosophical grounds for Michael Soulé's assertion that the untimely 'extinction of populations and species is bad'. After providing an explanation of the disciplinary perspective taken, the concept of extinction and its history within that literature are introduced. Then, different reasons for why anthropogenic extinctions might be morally problematic are presented based on (1) the loss of species' value, (2) harm to nonhuman individuals, (3) the loss of valuable biological variety and (4) duties to future generations. Subsequently, (5) cases where anthropogenic extinctions might be justified are considered. Regarding how to respond to extinctions, a selection of topics is addressed concerning (1) risks and (2) de-extinction. Finally, the review introduces other viewpoints from the 'extinction studies' literature.

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
