## [Reviewer Report]

Whether to publish this paper or not depends entirely on the goals of the journal.

This paper does an admirable job of surveying some of the current and historic literature in extinction ethics. It misses some of the nuances of the field, but that’s difficult to capture in a survey paper. That said, nothing in this paper is novel. The authors advance no new arguments, and do not critically engage with any of the work being done. Overall the paper is a useful tool for people looking to get a rough overview of the field in a western context, but doesn’t advance the conversation beyond this.

There are two significant amendments that I think need to be made before it could be published.

First, it would be useful to acknowledge the extensive environmental ethics that has been done in non-western contexts. This paper notes that it’s only looking at Anglophone literature, but more accurately it’s only considered traditional western environmental ethics. There’s a large number of non-western texts written in English, or that have now been translated, which are impactful on the field. A paragraph at least gesturing at what other work has been done outside of traditional western philosophy, and what sort of topics are covered would be good. Gary Snyder’s Practice of the Wild (1990) deals with species extinction and value from a native American perspective, for example.

Second, the paper would benefit from a slight restructuring. The discussion of risk in section 4 would be better put earlier in the paper, since it has themes which resonate with a number of other points. Instead, in Section 4, there ought to be an explicit section on artificial/assisted adaptation and migration, both of which are standard examples in the ethics literature of work done to mitigate and prevent extinction. While these topics are mentioned, they aren’t given the space they deserve in a review of extinction ethics.

A few other thoughts include that it might be worth mentioning the screw worm in the deliberate extinction section. There’s too much emphasis on diseases, which most people agree have a different set of ethical considerations to animals. Screw worms, which are actively being considered for deliberate extinction, would provide a stronger example of animal extinction alongside malaria-carrying mosquitoes.

I also recognise that while this paper has explicitly set-aside attempts to critically engage with the work being surveyed in this paper, it would be valuable in the conclusion to at least attempt to summarise any strengths or weaknesses observed. This would also increase the novelty of the paper beyond simple summary.

In all, the biggest weakness of this paper is that it doesn’t advance the conversation in any serious way. This is an overview, and if that’s the goal of the journal then it does an admirable job trying to cover significant ground.

---

## [Reviewer Report]

Fantastic overview of the Anglophone literature in environmental ethics that bears on efforts to achieve the deliberate extinction of a species. I learned much from it. The authors are right to underscore the complexity of the “concepts, arguments, and forms of analysis” relevant to these efforts.

One complexity overlooked here is what appears to be a widely accepted view among the public that the worth of different kinds of species varies depending roughly on where the species seems to fall in what theologians have sometimes called “the great chain of being.” For many people, the worth of a bacteria or virus species is intuitively less than that of an insect, and that of an insect less than that of a mammal. This may well be part of the reason that extinction of smallpox or polio does not seem concerning. So far as I am aware, this complexity is also overlooked in environmental ethics literature that directly discusses the ethics of extinction; however, it might have been to mentioned in the discussion “Understanding Extinction.” Indeed, among the terms whose definitions need examination is the very concept of “life,” since it is not entirely unclear what the status of viruses is.

Within the review of the environmental ethics literature, one point where I might take issue is over whether claims about the value of species quite captures the concerns that some environmental ethicists have expressed. I agree that this is the plausible place to start. Another way to approach environmental ethics and therefore the normative issues of deliberate extinction is through general principles or ideals about the human relationship to nature. The values-of-species approach typically starts with claims about the conditions that allow things to have value; the principle- or ideal-based approach asks more diffuse questions about moral attitudes toward nature in general or toward the human relationship to nature, or about human intervention into processes, laws, or general features of nature. In my reading, this is what Christopher Preston is thinking about when he laments human intrusion in “arrangements,” for example. It’s also one way of understanding Sylvan’s last-person-on-earth thought experiment: Sylvan can be understood as raising a very general question about “human chauvinism” and destruction. The questions at stake have to do, for example, with the desirability of human domination of nature or the importance of relations of respect or care for nature. I am not convinced that they can be rephrased strictly in terms of claims about the value that attaches to species; but if they can, the value of species is in some sense derivative from values found in other aspects of the human relationship to nature.

---

## [Editor Report]

I have received two reviews of this manuscript. Both reviewers engaged with the manuscript in detail and each provides fair and constructive comments that could yield improvements to the manuscript without requiring extensive editing. Given the nature of the comments, I don’t see need for further review but would like the authors to consider and respond to the comments. I agree with Reviewer #1 that the manuscript would benefit from some stance taken by the manuscript (and its authors), even if only in the concluding paragraphs.

---

## [Editor Report]

I am satisfied that the authors addressed the reviewer comments and do not see a need for further review. I have a few small items that the authors can address when they send back a final version of the manuscript.

15: “ask” should be “asks” 

41: I’ve usually seen this written as “pertaining to…”. I couldn’t quickly find an authoritative answer on the grammar, but please confirm this usage (here and elsewhere in the ms) is correct. 

87: “authors that” should be “authors who” 

122: I believe no-longer-existing should be hyphenated here. 

187: argumentative is redundant and can be deleted here. 

315: “Rolston’s” should be “Roltson” here. 

459-460: With use of “not” early in the sentence, I believe this should be followed by either/or rather than neither/nor. 

821: “that” should be “than”